# Heart Failure Promotes Cancer Progression in an Integrin β1-Dependent Manner

**DOI:** 10.3390/ijms242417367

**Published:** 2023-12-11

**Authors:** Irina Langier Goncalves, Lama Awwad, Sharon Aviram, Talel Izraeli, Laris Achlaug, Ami Aronheim

**Affiliations:** Department of Cell Biology and Cancer Science, Ruth and Bruce Rappaport Faculty of Medicine, Technion-Israel Institute of Technology, Haifa 3109601, Israel; irinalan@campus.technion.ac.il (I.L.G.); lamaaw@campus.technion.ac.il (L.A.); avirams@technion.ac.il (S.A.); talelm@campus.technion.ac.il (T.I.); laris1010.ab@gmail.com (L.A.)

**Keywords:** cardiac remodeling, cardiac dysfunction, cancer, Integrin β1, Periostin

## Abstract

Heart failure and cancer are currently the deadliest diseases in the Western world, posing the most pressing clinical challenges that remain unmet today. Both conditions share similar risk factors, including age, genetics, lifestyle, chronic inflammation, stress, and more. Furthermore, medications that are being used to counteract cancer frequently result in cardiotoxicity and the spontaneous emergence of heart failure. Thus, heart failure and cancer display an intimate connection and share similarities. Recent studies show that cardiac remodeling and heart failure promote cancer progression and metastasis. Using three different mouse models for heart failure revealed that the communication between the remodeled heart and the tumor is facilitated through multiple secreted factors. Among these factors, Periostin was consistently found to be elevated in all models and was shown to be required in vitro. Yet, whether Periostin is necessary for tumor promotion in vivo is unknown. Towards this end, we examined tumor promotion in mice lacking *Periostin* following transverse aortic constriction (TAC). Despite the loss of Periostin, tumor growth was promoted in the TAC-operated mice. This likely occurred due to increased levels of various cytokines and growth factors in *Periostin* KO mice. Many of these factors are potential ligands of Integrin receptors. Therefore, we next studied the role of Integrin receptors in the tumor-promotion phenotype following heart failure. We generated cancer cells with an *Integrin β1* loss of function mutation and examined tumor growth in the presence and absence of heart failure. *Integrin β1* KO cancer cells fail to display cardiac-remodeling-dependent tumor-promotion. Interestingly, a previous study showed that renal cell carcinoma cells (Renca) fail to be promoted following a myocardial infarction. Consistently, we show that Renca cells do not respond to secreted factors derived from the failing heart both in vitro and in vivo. Interestingly, Renca cells display low basal mRNA levels of *Integrin β1* which may explain the inability of heart failure to promote their growth. The findings may have significant clinical relevance to cardio–oncology patients who suffer from cancers with high levels of Integrin β1. Chemotherapy leading to cardiotoxicity in these patients may generate a vicious cycle with poor prognosis.

## 1. Introduction

Cardiac hypertrophy is associated with clinically significant conditions including high blood pressure, coronary artery disease, valvular disorders, and obesity-related cardiomyopathy [1,2,3]. Advances in the treatment of cardiovascular diseases such as: lowering risk factors, improved control of blood pressure, and better treatment modalities in valvular replacement surgery have led to a reduction in heart failure and increased lifespan. The increased lifespan exposes cardiovascular patients to diseases associated with aging such as cancer [4]. Notably, cardiac growth has long been thought of as a “tumor-like” growth. The biological principles of cell growth, death, and survival are as important in the onset of heart failure as they are in tumor progression [5,6]. Moreover, cardiac diseases and cancer have common risk factors, including genetic predisposition, smoking, obesity, hyperlipidemia, sedentary lifestyle, diabetes, aging, and more [7]. While heart failure and cancer have been considered separate diseases, it is becoming evident that these two maladies are highly connected and affect each other’s outcomes at multiple levels [8,9]. Despite many advances in treating both cardiac diseases and cancer, they have remained the leading causes of death.

A notable link between heart failure and cancer lies in the cardiotoxicity induced by various cancer treatments [10]. Additionally, multiple epidemiological studies suggest that patients with cardiovascular complications are at higher risk of developing cancer with poorer outcomes [11,12,13,14]. Significantly, the interaction between the heart and the tumor, and potentially with other organs in the body, takes place through immune cells and secreted factors [11,15]. Together, they determine the outcome for both cancer and cardiac disease. Interestingly, studies have shown that a mouse model for heart failure of myocardial infarction (MI) increased tumor load [11,16]. In those studies, different secreted factors seemingly played a crucial role in this promotion.

A study from our lab addressed the question if cardiac-remodeling in the absence of heart failure is sufficient to affect cancer severity and outcome [13]. In this study, we used a model of transverse aortic constriction (TAC), as a mouse model for early cardiac-remodeling processes. TAC-operated mice developed larger primary tumors with higher proliferation rates and displayed more metastatic lesions compared with the non-operated controls. Importantly, the serum derived from the TAC-operated mice potentiated cancer cell proliferation in vitro, suggesting the existence of secreted tumor-promoting factors. RNA-seq data showed a significant elevation of *Periostin* mRNA levels in the hearts of TAC-operated mice. Increased Periostin levels were also found in the serum following the TAC operation. Notably, the removal of Periostin from the serum halted the proliferation of cancer cells in vitro [13]. Conversely, the introduction of Periostin boosted cancer cell proliferation in vitro which is consistent with its well-established role in cell proliferation [17,18]. Two additional cardiac hypertrophy models: a genetically modified mouse model overexpressing the activating transcription factor 3 (ATF3) in cardiomyocytes [19], and a phenylephrine (PE) infusion representing a hypertension mouse model [20], were used to study tumor growth in our lab. These studies demonstrated that elevated levels of Periostin coincided with increased expression of additional cytokines, which may provide an explanation for the tumor-promoted growth observed in hypertrophied heart models.

The signaling pathways associated with tumor growth and cell proliferation are altered following the secretion of various growth factors in response to cardiac remodeling and heart failure. These factors include secreted extracellular matrix (ECM) proteins such as Fibronectin and Periostin, as well as various other cytokines [13,19]. The cancer cells’ capacity to receive and interpret signals from these secreted factors relies on the existence of specific membrane receptors on their cell surface. These include Integrins, which are present in various cells, including cancer cells [21]. In this study, we initially examined the role of Periostin in cardiac-remodeling-dependent tumor-promotion using *Periostin* knockout mice (KO). Surprisingly, tumor promotion in response to cardiac remodeling was preserved in these mice. We found that *Periostin* KO mice compensate for the loss of Periostin by increasing the expression of multiple secreted factors that are known to associate with Integrin receptors. Correspondingly, Integrin β1 receptor ablation on cancer cells resulted in a loss of the tumor-promotion phenotype following cardiac remodeling. Interestingly, cancer cells such as renal carcinoma cancer cells (Renca) are not promoted in mice with heart failure that display low basal mRNA levels of *Integrin β1*.

Collectively, the Integrin β1 receptors expressed on the cell membranes plays a pivotal role in the cardiac-remodeling-dependent tumor-promotion phenotype, suggesting a potential therapeutic target. This implies that targeting Integrin β1 is a potential strategy for developing therapeutic interventions to suppress the cardiac-remodeling tumor-promotion vicious cycle.

## 2. Results

### 2.1. Cardiac Remodeling following TAC Promotes Tumor Growth in the Periostin Knockout Mice Strain

Based on our previous study [13], we hypothesized that tumor promotion following cardiac remodeling is dependent on Periostin expression. To generate a mouse model with an inactivation mutation in the *Periostin* gene, we took advantage of the *Periostin*-CRE-ER knock-in mice and crossed the mice to obtain CRE-ER homozygous mice leading to a *Periostin* gene inactivation mutation (POSTN^(−/−)^). To validate the POSTN^(−/−)^ mice, we first tested the Periostin levels in the serum derived from C57Bl/6 and POSTN^(−/−)^ mice. Whereas Periostin levels found in C57Bl/6 mice are relatively high (2544 ng/mL), the levels were under the detectable levels in the serum of POSTN^(−/−)^ mice (Figure 1A).

Next, to examine how the loss of Periostin affects the promotion of tumor growth following cardiac remodeling, LLC cells were subcutaneously implanted into the flanks of male POSTN^(−/−)^ mice. Tumor growth was monitored in the TAC-operated and the non-operated control group (Figure 1B). The TAC-operated POSTN^(−/−)^ group displayed lower contractile function than the non-operated group, with an FS of ~24% compared to ~29%, respectively (Appendix A). Cardiac dysfunction was accompanied by heart hypertrophy, as the ventricular weight to body weight ratio (VW/BW) was significantly higher in the TAC-operated POSTN^(−/−)^ group (Appendix A). These results are consistent with the previous findings from our lab for the TAC-operated tumor-bearing C57Bl/6 mouse strain [13]. Although no Periostin is detected in the serum derived from these mice, tumor growth that was monitored over time showed higher tumor volume in the TAC-operated POSTN^(−/−)^ group as compared to the non-operated control (Figure 1C). In addition, tumors from TAC-operated mice were heavier at the endpoint (Figure 1D).

### 2.2. Tumor Promotion Following Cardiac Remodeling in the POSTN^(−/−)^ Mice Is Independent of Periostin Expression

The tumor-promotion phenotype was preserved in TAC-operated POSTN^(−/−)^ mice despite the loss of *Periostin* expression. To study this, we conducted a proliferation assay in vitro with serum derived from the POSTN^(−/−)^ serum. Serum was collected from either naïve C57Bl/6 or POSTN^(−/−)^ mice and used to supplement the serum-free culture medium of growing LLC and polyoma middle T breast cancer cells (PyMT) cells (2% mouse serum). Both cell lines displayed higher cell proliferation when grown in the POSTN^(−/−)^ mice serum compared to the serum derived from control C57Bl/6 mice (Figure 2A,B).

Then, we examined whether serum derived from POSTN^(−/−)^ mice contained higher levels of growth factors. Towards this end, we used a proteome cytokine array probed with serum derived from POSTN^(−/−)^ compared to C57Bl/6 mice. POSTN^(−/−)^ mice serum displays higher levels of pro-inflammatory and pro-tumorigenic cytokines as compared to control C57Bl/6 mice serum (Figure 2C). Periostin is a known component of the ECM proteins, thus its absence may lead to compensation with similar ECM proteins. Indeed, ELISA analysis for Fibronectin, another ECM component, is expressed twice as much in POSTN^(−/−)^ mice serum as compared to control C57Bl/6 mice serum (Figure 2D). Collectively, the data suggests that loss in Periostin leads to alteration in the levels of multiple secreted proteins that compensate and mediate the tumor-promotion phenotype following TAC in POSTN^(−/−)^ mice.

### 2.3. Integrin β1 Is Required for Mediating the Connection between the Remodeled Heart and Cancer Cell Promotion

One of the major cell surface receptors for Periostin, Fibronectin, and other ECM proteins are Integrins [22]. Integrin β1 specifically plays a pivotal role in the binding of multiple ECM proteins [23]. To examine the role of Integrin β1 in mediating the connection between the remodeled heart and the tumor, we have generated a PyMT breast cancer cell line with a loss of function mutation of *Integrin β1* (PyMT ITGB1 KO) (Appendix A). To examine the growth of ITGB1 KO cells in vitro, we compared their growth to the PyMT parental cells in the absence or presence of the 2% serum. While no difference was observed when grown in the presence of 2% FBS, significantly lower growth was observed when grown in the presence of C57Bl/6 mice serum (Figure 3A). Next, to examine whether cardiac remodeling promotes ITGB1 KO tumor growth, we implanted PyMT and ITGB1 KO breast cancer cells into the mammary fat pad of female C57Bl/6 mice. Two days later, each cohort was divided into TAC-operated, and non-operated control groups (Figure 3B). In the PyMT WT group, tumor volume was significantly higher in the TAC-operated group as compared to the non-operated control group (Figure 3C). Similarly, tumor weight at the endpoint was heavier in the TAC-operated mice group (Figure 3D). In contrast, in the group that was implanted with PyMT ITGB1 KO cells, there was no significant difference between the TAC-operated and the non-operated mice groups in both tumor volume (Figure 3C) and tumor weight at the endpoint (Figure 3D).

The ©nability to promote cancer cell growth was not due to the difference in the extent of cardiac remodeling since TAC-operated mice in both experimental groups displayed significant cardiac remodeling, as exemplified by the low FS% and increased VW/BW ratio (Appendix A). Additionally, to address the slow proliferation rate of ITGB1 KO cells, we extended the duration of the experiment until the volume of ITGB1 KO tumors matched that of PyMT WT tumors at the endpoint. Thus, we conclude that Integrin β1 KO in PyMT breast cancer cells abrogates the tumor-promotion phenotype following TAC.

A previous study showed that myocardial infarction (MI) failed to promote tumor growth of the renal murine cell line (Renca) [24]. Nevertheless, no possible explanation for the observed phenotype was provided. Therefore, we sought to examine whether the Renca cell line fails to increase tumor growth following TAC operation as well. BALB/c male mice were subcutaneously implanted into the flanks with Renca cells followed by a TAC operation (Figure 4A). Although TAC-operated mice displayed lower FS% and an increased VW/BW ratio (Appendix A), the volume of Renca-cells-derived tumors did not show any significant difference in volume and weight (Figure 4B,C).

Renca cells fail to promote growth in response to cardiac remodeling in vivo. We next examined how Renca cells respond to the serum derived from TAC-operated mice in vitro. Unlike PyMT and LLC cells that show increased cell proliferation in the presence of the TAC-operated serum, ITGB1 KO and Renca cells failed to display increased cell proliferation (Figure 5A). We next examined whether the mRNA level of *Integrin β1* in Renca cells may explain the inability to respond to cardiac remodeling. Interestingly, the basal mRNA levels of *Integrin β1* in Renca cells are significantly lower as compared with PyMT cells and are comparable to PyMT ITGB1 KO cells (Figure 5B). In the tumor stroma, the mRNA levels of *Integrin β1* are not changed following TAC and are significantly lower in the tumors derived from both ITGB1 KO and Renca cancer cells (Figure 5C). These findings further support that Integrin β1 plays a central role in the promotion of tumor growth following TAC.

A schematic summary of the main manuscript findings and conclusions is provided in Figure 6. Collectively, our findings suggest that *Periostin* ablation was not sufficient to abrogate the promotion of tumor growth following cardiac remodeling, due to the elevation of multiple growth factors that include pro-inflammatory and tumorigenic cytokines and Fibronectin. Yet, *Integrin β1* KO in cancer cells is sufficient to abolish the tumor-promotion phenotype in TAC-operated mice. Low basal mRNA levels of *Integrin β1* in Renca cancer cells may explain the lack of tumor-promotion phenotype following TAC. Therefore, the levels of Integrin β1 in cell lines may be a key player in communication between the remodeled heart and the tumor.

## 3. Discussion

In recent years, a growing body of evidence has demonstrated a link between cardiovascular diseases and tumor growth in murine models [11,13,19,20] and humans [7,14]. Revealing and comprehending the mechanism through which tumor promotion occurs is important. Cancer treatments can lead to cardiovascular complications, initiating a vicious cycle that may further promote tumor growth [15]. In this context, we investigated the necessity of Periostin, an extracellular matrix (ECM) protein associated with tumor promotion [13,17,18,19,20]. Using a mouse strain lacking Periostin, we failed to demonstrate the absolute requirement of Periostin to the observed tumor-promotion phenotype. This is due to a compensatory mechanism for the lack of Periostin during development via alteration of expression of multiple pro-inflammatory cytokines and growth factors. Previously, it was shown that Periostin KO mice had no abnormalities in the morphology of cardiomyocytes [25] nor alteration in the ventricular weight [26]; however, those mice had dramatic alteration of gene expression of cardiac fibroblast. It was shown that cardiac fibroblast from Periostin KO mice had different molecular programming, probably due to a lack of Periostin during embryonic development [26]. In addition, Periostin KO mice are prone to ventricular rupture after MI or TAC, nevertheless, the surviving mice display significantly lower fibrosis and better cardiac function [25,26]. Moreover, cells expressing Periostin are identified as perivascular. Consequently, it is crucial to take into account the influence of aging on both the vasculature and perivascular niches. Aging has been demonstrated to affect perivascular microenvironments, and gaining insights into how these alterations might impact the study’s outcomes would markedly enhance the overall comprehension of the research [27]. Combining these observations with our findings reveals that the knockout of *Periostin* results in a comprehensive modification in gene expression, influencing fibroblasts, cardiomyocytes, and various other cell functions. Those findings together with ours, demonstrate that *Periostin* KO leads to global alteration in gene expression that changes fibroblast, cardiomyocytes, and other cell functions.

Since tumor promotion was found to be mediated by multiple cytokines and growth factors involving various cell surface membrane receptors found on cancer cells, we were interested in examining whether a central Integrin receptor may play a key role in mediating the ECM signaling leading to tumor promotion. Integrin receptors are composed of multiple subunits. The key player in association with Integrin ligands is the Integrin β1 subunit [28]. Integrin β1 is the most abundantly expressed among Integrin β subunits found in almost all cell types [21] and is associated with at least 10 different Integrin α subunits [28]. Integrin β1 is a part of the receptor that binds to Periostin and Fibronectin, which are highly elevated following cardiac-remodeling processes [13,19,20]. Furthermore, Fibronectin was shown to be elevated in the serum of POSTN(−/−) mice, most likely as compensation for the absence of Periostin. Integrin receptors are known to be crucial for the growth, proliferation, and invasion of cancer cells [29]. Therefore, we have generated PyMT cancer cells with a loss of function mutation in the *Integrin β1* gene (PyMT ITGB1 KO). We show that PyMT cells lacking *Integrin β1* display normal growth in vitro when supplemented with FBS and slower growth in the presence of mouse serum compared to PyMT parental cells. PyMT ITGB1 KO cells when implanted into mice developed tumors at a slower rate. Importantly, the TAC operation failed to promote the growth of these cells as compared to the PyMT parental cell.

A previous study described how MI fails to promote the growth of tumors derived from the Renca cancer cell line [24]. The molecular mechanism involved in the lack of a tumor-promotion phenotype was not described. In this study, we confirmed that TAC fails to promote Renca cell growth. The lack of tumor promotion following cardiac promotion is consistent with the notion that Renca cells display low levels of *Integrin β1* and thus provides a possible explanation for the lack of tumor promotion.

Further research is needed to identify additional cancer cell lines lacking a tumor-promotion phenotype in response to heart failure. These studies will shed light on the importance of Integrin receptors’ expression on cancer cells and potentially provide a therapeutic approach to prevent chemotherapy leading to heart failure resulting in a tumor-promotion vicious cycle.

## 4. Materials and Methods

All experimental protocols were approved by the Institutional Committee for Animal Care and Use at the Technion, Israel Institute of Technology, Faculty of Medicine, Haifa, Israel. Approval number IL-069-05-20, IL-157-10-20-21. All study procedures comply with the NIH Guide for the Care and Use of Laboratory Animals guidelines. All experiments were randomized and blindly performed.

### 4.1. Cell Culture

The polyoma middle T (PyMT) murine breast carcinoma cells were derived from primary tumor-bearing transgenic mice expressing polyoma middle T under the control of the murine mammary tumor virus promoter [30]. PyMT cells were kindly provided by Prof. Tsonwin Hai (Ohio State University, Columbus, OH, USA). The LLC and Renca cancer cell lines were purchased from the American Type Culture Collection, ATCC. The PyMT *Integrin β1* KO cell line was generated by genome engineering using the CRISPR-Cas9 system [31]. All cell lines were tested by IDEXX BioAnalytics and found to be free of mycoplasma and viral contamination. Cells were cultured in DMEM/RPMI containing 10% FBS, 1% streptomycin and penicillin, 1% L-glutamine, and 1% sodium pyruvate (full medium) at 37 °C in a humidified atmosphere containing 5% CO_2_. Cancer cell implantation is used at maximal passage number five.

### 4.2. Generation of ITGB1-KO Cells

The sgRNA was designed according to exon 2 of the mouse *Integrin β1* gene and predicted by using the CRISPR track of the USCS genome browser (https://genome.ucsc.edu/, accessed on 11 May 2020) (Appendix A).

ITGB1 sgRNA: AAGCAGGGCCAAATTGTGGGTGG, was cloned into pSpCas9(BB)-2A-GFP(PX458) (Addgene plasmid #48138) according to the protocol of the Zhang laboratory (Broad Institute of Massachusetts Institute of Technology and Harvard University, Cambridge, MA, USA) [31] to obtain pSpCas9-ITGB1-GFP.

### 4.3. Cell Culture and Transfections

PyMT cells were transfected with pSpCas9-ITGB1-GFP using the PolyJet (SingaGen, Rockville, MD, USA; Cat# SL100688) transfection reagent according to the manufacturer’s instructions.

Cell culture medium was replaced with fresh medium 24 h post-transfection, and cells were harvested 48 h thereafter for GFP sorting. Single-cell clones of GFP positive were isolated by a cell sorter (BD FACSMelody, BD Biosciences, Franklin Lakes, NJ, USA; FacsChorus version 1.4.3.0) in 96-well plates (doublet discriminated; FSC and SSC). ITGB1 expression of expanded colonies was determined by Western blot, according to the protocol as described in our previous paper by our lab [32]. Clones showing complete loss of ITGB1 were further confirmed by Sanger sequencing of exon 2 of ITGB1.

### 4.4. Antibodies

The primary antibodies used were anti-ITGB1 (Cat# 374429), and anti-GAPDH (Cat# 25778) which were purchased from Santa Cruz (Santa Cruz, CA, USA).

### 4.5. Cell Proliferation Assay

Cancer cells were seeded in full medium (10% FBS) at a concentration of 5 × 103 cells/mL for 6 h. After the cells were attached to the plate, the medium was replaced with a serum-free medium overnight. Then, the medium was replaced with the fresh medium containing 2% (FBS; positive control), serum-free (negative control), or 2% mouse serum derived from naïve or TAC-operated mice and 1% streptomycin and penicillin, and 1% L-glutamine DMEM/RPMI medium. Cancer cells were left with FBS or mouse blood serum for 24, 48, and 72 h. The proliferation rate was measured with the CellTiter-Glo luminescent cell viability assay according to the manufacturer’s instructions. Luciferase activity was measured with a TD 20/20 luminometer (Turner Designs, Sunnyvale, CA, USA).

### 4.6. Animals

C57Bl/6C57BL/6, BALB/c (BALB/cJ The Jackson Laboratories Stock No: 000651) and heterozygote *Periostin*-CRE-ER (Postn-CRE-ER The Jackson Laboratories Stock No: 029645) were purchased from The Jackson Laboratory (Bar Harbor, ME, USA). The heterozygote *Periostin*-CRE-ER mice were bred to obtain a homozygote Periostin knockout strain (POSTN^(−/−)^). Mice were bred and raised at the Pre-Clinical Research Authority at the Ruth and Bruce Rappaport Faculty of Medicine. Mating cages were maintained under regular chow and water. Mice were weaned into separate cages at three weeks of age. Surgery and sacrificing procedures were carried out under isoflurane anesthesia. The number of mice used in each experiment is indicated in the figure legends.

### 4.7. Cancer Cells Implantation

PyMT (WT or ITGB1 KO) cancer cells were orthotopically injected into the mammary fat pad of female 8–10-week-old mice (0.5 × 106 cells per mouse) and into the back flanks of male 8–10-week-old mice. LLC and Renca cancer cells were implanted (0.5 × 106 per mouse) into the back flanks of mice. Tumor size was measured using a caliper, and tumor volume was calculated with the formula: Width × Length × 0.5. The humane endpoint is defined when the tumor size reaches 1500 mm^3^, according to the Institutional Animal Care and Use Committee.

### 4.8. Transverse Aortic Constriction

Transverse aortic constriction (TAC) surgery was performed on 8–10-week-old male mice. Constriction was performed using a 27G blunt needle to create a standardized constriction of the aorta as was previously described [33]. All TAC procedures in this study were performed by the same individual, blinded to the mice genotype.

### 4.9. Echocardiography

Mice were anesthetized with 1% isoflurane and kept on a 37 °C heated plate throughout the procedure. Echocardiography was performed with the Vevo3100 micro-ultrasound imaging system (VisualSonics, Fujifilm, Toronto, ON, Canada) equipped with 13- to 38-MHz (MS 400) and 22- to 55-MHz (MS550D) linear array transducers. Cardiac size, shape, and function were analyzed using conventional two-dimensional imaging and M-mode recordings. Maximal left ventricular end-diastolic (LVDd) and end-systolic (LVDs) dimensions were measured in short-axis M-mode images. Fractional shortening (FS) was calculated with the following formula: FS% = [(LVDd − LVDs)/LVIDd] × 100. FS value is based on the average of at least three measurements for each mouse.

### 4.10. RNA Extraction

The cells’ culture RNA extraction was performed using the NucleoSpin RNA kit (740984.50; Macherey Nagel, Düren, Germany) following the manufacturer’s instructions.

mRNA was extracted from hearts and tumors using an Aurum total mRNA fatty or fibrous tissue kit (no. 732–6830, Bio-Rad, Hercules, CA, USA) according to the manufacturer’s instructions. Next, cDNA was synthesized from 1000 ng purified mRNA (from tissue or culture) with the iScript cDNA synthesis kit (no. 170–8891, Bio-Rad) according to the manufacturer’s instructions.

### 4.11. Quantitative Real-Time PCR

Quantitative real-time polymerase chain reaction (qRT-PCR) was performed with QuantStudio3 (Thermofisher Scientific, Carlsbad, CA, USA). Serial dilutions of a standard sample were included for each gene to generate a standard curve. Values were normalized to GAPDH and βactin expression levels for the in vitro and in vivo cancer cells and tumor tissues, respectively.

### 4.12. Blood Serum

Blood was obtained from the facial vein with a 4 μm sterile Goldenrod animal lancet (MEDIpoint, Inc., Mineola, NY, USA). Blood was collected and allowed to clot at room temperature for 2 h, followed by 15 min of centrifugation at 2000× *g*. The serum was immediately aliquoted and stored at −20 °C for future use.

### 4.13. ELISA

The quantification of Periostin and Fibronectin in the serum was performed using the mouse Periostin/OSF-2 Quantikine ELISA kit (R&D systems Inc., Minneapolis, MN, USA) and mouse Fibronectin ELISA kit (E-EL-M0506, Elabscience, Houston, TX, USA), according to the manufacturer’s instructions.

### 4.14. Cytokine Array

Serum was collected from naïve C57BL/6C57BL/6 and POSTN^(−/−)^ (*n* = 3 per group) and was pooled and applied (75 µL) to Proteome Mouse XL Cytokine Array filters (Cat. ARY028, R&D Systems, Norcross, GA, USA) according to the manufacturer’s instructions. The signal corresponding to each factor in the array was quantified by TotalLab software analysis (Version 2.2). The level of expression of each protein on the array was calculated relative to the value obtained for the positive control.

### 4.15. Statistical Analysis

Data are presented as mean ± SE. All mice were included in each statistical analysis unless they were euthanized for humane reasons before the experimental endpoint. Experimental groups were blinded to the experimentalists during data collection. Animals were selected for each group in a randomized fashion. The number of mice in each experimental group was at least *n* = 3. The statistical significance of tumor volume was determined by two-way repeated-measures ANOVA followed by the Bonferroni post-test. Comparison between several means was analyzed by one-way ANOVA followed by Tukey’s post-test. Comparison between two means was performed by two-tailed Student’s *t*-test or Mann–Whitney U test. Analyses were performed with GraphPad Prism 9 software. Values of *p* < 0.05 were accepted as statistically significant.

## Figures and Tables

**Figure 1 ijms-24-17367-f001:**
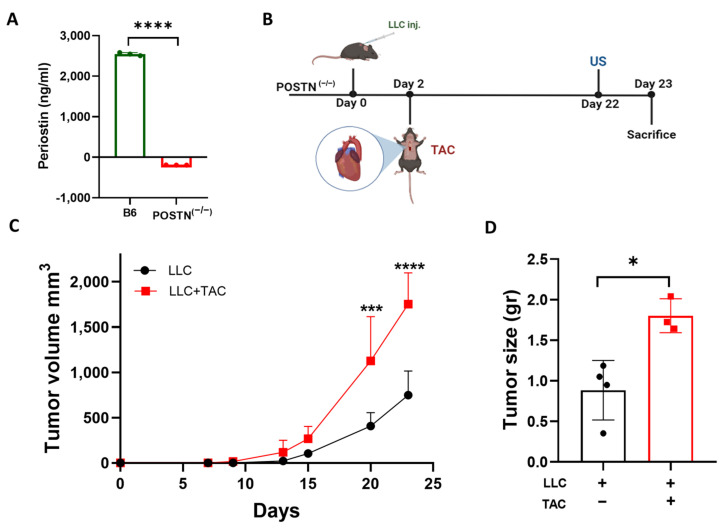
TAC-operation promotes LLC cancer cell growth in periostin KO mice. (**A**) Serum levels of B6 (green) and POSTN^(−/−)^ (red) mice (*n* = 3 per group) were obtained by ELISA for Periostin. (**B**) Schematic experimental timeline for the Lewis lung carcinoma (LLC) cell model implanted in Periostin KO (POSTN^(−/−)^) male mice. (**C**) POSTN^(−/−)^ mice were subcutaneously implanted into the flanks with LLC cells (0.5 × 106 cells per mouse) (*n* = 7). Mice were divided into 2 groups: TAC-operated group (red, *n* = 3), and non-opereted group (black, *n* = 4). Tumors were monitored over time and tumor volume was calculated using the formula: Width × Length × 0.5. (**D**) Tumor weight at sacrifice. Each dot represents one mouse. Data are presented as mean ± SE. Two-way repeated measures ANOVA followed by the Bonferroni post-test (**C**), and Student’s *t*-test (**A**,**D**). * *p* < 0.05, *** *p* < 0.001, **** *p* < 0.0001.

**Figure 2 ijms-24-17367-f002:**
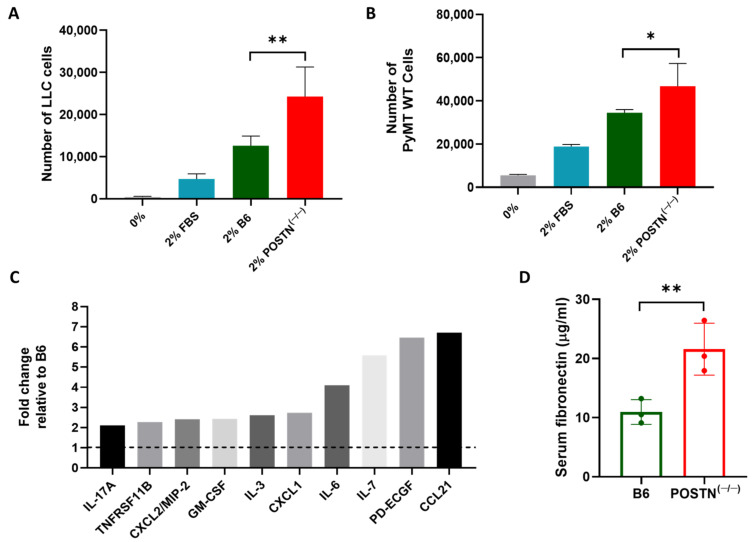
Tumor-promotion phenotype in the serum derived from POSTN^(−/−)^ mice is mediated by multiple secreted factors. The proliferation of (**A**) polyoma middle T breast cancer cells (PyMT) or (**B**) Lewis lung carcinoma (LLC) cells under different growth conditions in the presence of the indicated murine serums; *n* = 4 plates per treatment for each time point. (**C**) Pooled serum from POSTN^(−/−)^ or C57Bl/6 male mice (*n* = 3 per pool) was used to probe the proteome cytokine array. For each protein, the serum levels are presented as a fold change in the serum derived from POSTN^(−/−)^ relative to the control C57Bl/6 serum. (**D**) Fibronectin (FN) serum levels measured by ELISA (*n* = 3 per group; C57Bl/6 in green, POSTN^(−/−)^ in red). Data are presented as mean ± SE. One-way ANOVA followed by Tukey’s post-test (**A**,**B**), and Student’s *t*-test (**D**). * *p* < 0.05, ** *p* < 0.01.

**Figure 3 ijms-24-17367-f003:**
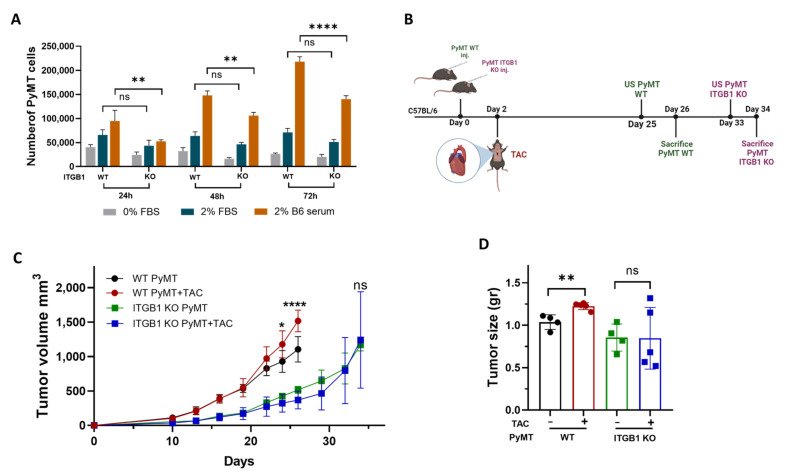
*Integrin β1* KO PyMT cancer cells abrogate the tumor-promotion phenotype following TAC. (**A**) Polyoma middle T WT (PyMT WT) and ITGB1 KO (PyMT ITGB1 KO) cells were cultured in the absence of the serum (0%) or the presence of 2% FBS, or 2% C57Bl/6 mice serum for 24, 48, and 72 h. Cell proliferation was measured with the luminescent cell viability assay; *n* = 4 plates per treatment for each time point. (**B**) Schematic experimental timeline for PyMT (WT/ITGB1 KO) cells implanted in TAC-operated and control C57Bl/6 female mice (**C**) PyMT (WT/ITGB1 KO) (0.5 × 106 cells per mouse) cells were orthotopically injected into the back left side mammary fat pad of C57Bl/6 mice (WT PyMT; black, *n* =3, WT PyMT+TAC; red, *n* = 5, PyMT ITGB1 KO; green, *n* = 3, PyMT ITGB1 KO + TAC; blue, *n* = 5). Tumors were monitored over time and tumor volume was calculated using the following formula: Width × Length × 0.5. (**D**) Tumor weight at sacrifice. Each dot represents one mouse. Data are presented as mean ± SE. Two-way repeated measures ANOVA followed by the Bonferroni post-test (**A**,**C**), and one-way ANOVA followed by Tukey’s post-test (**D**). ns—not significant, * *p* < 0.05, ** *p* < 0.01, **** *p* < 0.0001.

**Figure 4 ijms-24-17367-f004:**
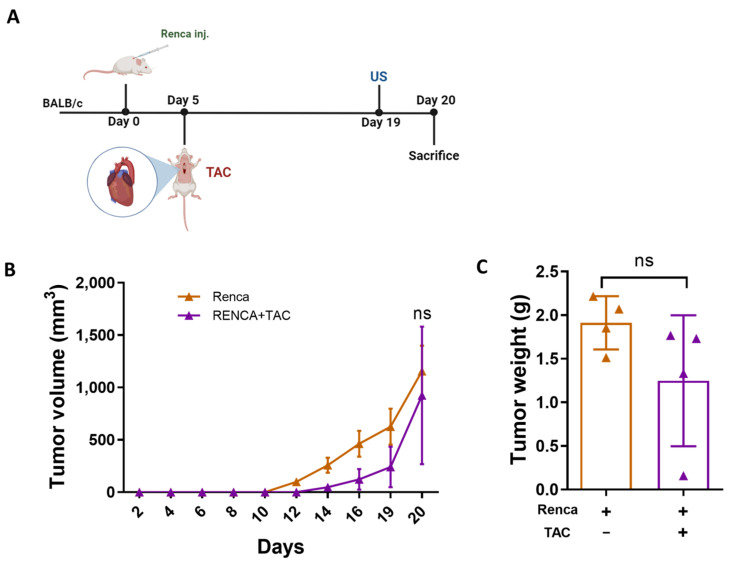
TAC-operation fails to promote Renca cancer cells’ tumor growth. (**A**) Schematic experimental timeline for Renca cell model of BALB/c female mice TAC-operated vs. non-operated control. (**B**) BALB/c female mice (*n* = 8) were orthotopically injected into the back left side mammary fat pad with Renca cells (0.5 × 106 cells per mouse). Mice were divided into 2 groups: TAC-operated group (purple, *n* = 4), and non-operated group (orange, *n* = 4). Tumors were monitored over time and tumor volume was calculated using the following formula: Width × Length × 0.5. (**C**) Tumor weight at sacrifice. Each dot represents one mouse. Data are presented as mean ± SE. Two-way repeated measures ANOVA followed by the Bonferroni post-test (**B**) and Student’s *t*-test (**C**). ns—not significant.

**Figure 5 ijms-24-17367-f005:**
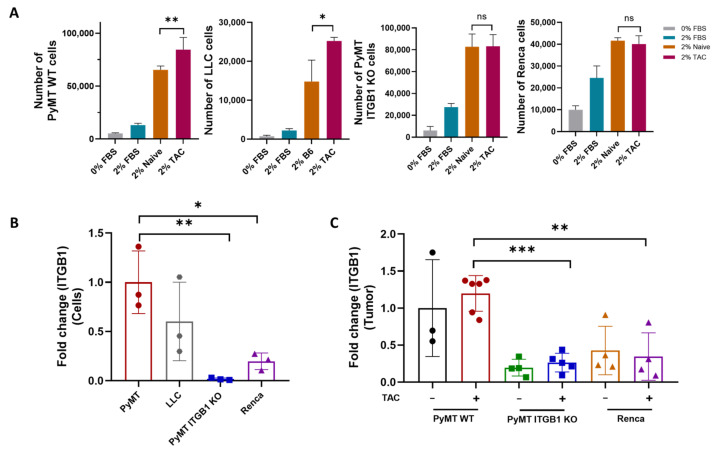
The tumor-promotion phenotype following TAC is Integrin-β1-dependent. (**A**) The growth of the indicated cancer cell lines was examined 48 h in culture in the absence of serum (0%) or the presence of 2% FBS, or 2% mouse serum derived from non-operated or TAC-operated mice. Cell growth was measured by the luminescent cell viability assay; *n* = 4 plates per treatment. (**B**) mRNA was extracted from the indicated cancer cells (**A**), cDNA was prepared and the mRNA expression level of the *Integrin β1* was analyzed by qRT-PCR. The GAPDH housekeeping gene was used to normalize mRNA levels (*n* = 3 repeats per cancer line). (**C**) mRNA was extracted from tumors (PyMT WT, LLC, PyMT ITGB1 KO, and Renca) of non-operated or TAC-operated mice. cDNA was prepared and the mRNA expression level of the *Integrin β1* was analyzed by qRT-PCR. βACTIN housekeeping gene was used to normalize mRNA levels. Each dot represents one mouse. Data are presented as mean ± SE. One-way ANOVA followed by Tukey’s post-test (**A**–**C**). ns—not significant, * *p* < 0.05. ** *p* < 0.01, *** *p* < 0.001.

**Figure 6 ijms-24-17367-f006:**
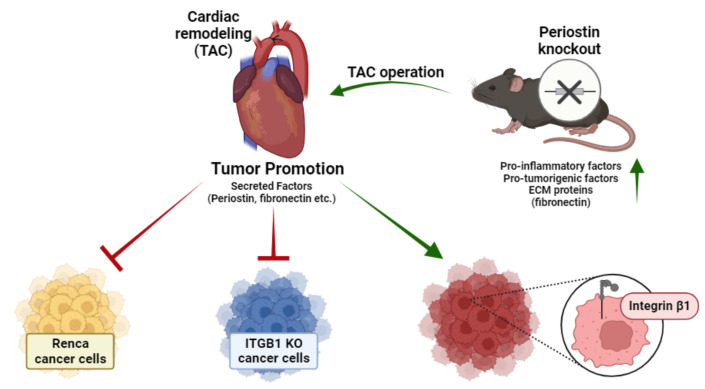
Graphical abstract describing the manuscript’s main findings. Cardiac remodeling promotes tumor growth and proliferation via secreted factors (even in the POSTN^(−/−)^ mice). Integrin β1 is the key player found at the receiving end of those circulating signals on the cancer cell membrane. Promotion is represented as a green arrow line and suppression is represented as a red inhibition arc.

## Data Availability

All the obtained data used to support the findings of this study are available from the corresponding author upon reasonable request.

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
