# Peer review of "Heart Failure Promotes Cancer Progression in an Integrin β1-Dependent Manner"

_ijms, 2023, doi:10.3390/ijms242417367_

Round 1
Reviewer 1 Report
Comments and Suggestions for Authors
This paper investigates how the stress of cardiac diseases impacts the outcome of cancer treatments. A few points I would like to have answered:
Periostin mice: are the mice purchased from Jackson Laboratory hetero- or homozygous for periostin? If not, with which mice were the periostin mice bred to obtain heterozygote mice?
Paragraph 4.5. about the Cell Proliferation Assay says that cancer cells were seeded in “full medium”. Please indicate what “full medium” is.
In paragraph 4.7 on page 13 the authors outline their calculation of tumor volume. Normally volume = length x width x height. I don’t really understand the formula the authors are using (width2 x length x 0.5). From where is this formula derived? The legend for figure 1B uses width X length X 0.5. Which formula is used and from where is 0.5 coming from?
Results:
Since the supplemental figures 1A-1C data characterize the animal model, they should be presented in the main manuscript.
Figure 2E shows no error bars. Does this experiment represent n=1?This paper investigates how the stress of cardiac diseases impacts the outcome of cancer treatments.
Author Response
This paper investigates how the stress of cardiac diseases impacts the outcome of cancer treatments. A few points I would like to have answered:
- Periostin mice: are the mice purchased from Jackson Laboratory hetero- or homozygous for periostin? If not, with which mice were the periostin mice bred to obtain heterozygote mice?
Response: Thank you for the comment we noted and resolved it. Heterozygotes to periostin CRE-ER from Jackson. We added this data to the method section.
- Paragraph 4.5. about the Cell Proliferation Assay says that cancer cells were seeded in “full medium”. Please indicate what “full medium” is.
Response: Thank you for the comment we noted and resolved it. 10% FBS is the full medium. We added this data to the method section.
- In paragraph 4.7 on page 13 the authors outline their calculation of tumor volume. Normally volume = length x width x height. I don’t really understand the formula the authors are using (width2x length x 0.5). From where is this formula derived? The legend for figure 1B uses width X length X 0.5. Which formula is used and from where is 0.5 coming from?
Response: Thank you for the comment we noted and resolved it. This is a formula for estimating tumor volume using the 0.5 x L x W x W. It is a common and simplified approach used in clinical practice and research, and it assumes that the shape of the tumor is an Ellipsoid. This simplified formula is easier to use and is often considered a reasonable approximation for estimating tumor volume in clinical settings. See references below:
- Sreenivasan, S. A., Madhugiri, V. S., Sasidharan, G. M., & Kumar, R. V. (2016). Measuring glioma volumes: A comparison of linear measurement based formulae with the manual image segmentation technique. Journal of Cancer Research and Therapeutics, 12(1), 161-168.
- Iliadis, G., Selviaridis, P., Kalogera-Fountzila, A., Fragkoulidi, A., Baltas, D., Tselis, N., ... & Fountzilas, G. (2009). The importance of tumor volume in the prognosis of patients with glioblastoma. Strahlentherapie und Onkologie, 185(11), 743.
Results:
- Since the supplemental figures 1A-1C data characterize the animal model, they should be presented in the main manuscript.
Response: Thank you for the comment we noted and resolved it. Supplementary Figure 1A was moved to the main text. However, the remaining 2 graphs do not describe the POSTN(-/) strain. There is no comparison between the KO strain and WT mice in these graphs.
- Figure 2E shows no error bars. Does this experiment represent n=1? This paper investigates how the stress of cardiac diseases impacts the outcome of cancer treatments.
Response: Thank you for the comment we noted and resolved it. Serum was collected from naïve C57BL/6C57BL/6 and POSTN(-/-) was applied (75 µL) to Proteome Mouse XL Cytokine Array – the 75 µL is a pool of 3 different mice from each strain. We added this data to the method section.

Reviewer 2 Report
Comments and Suggestions for Authors
Heart failure and cancer are two of the deadliest diseases in the Western world, sharing common risk factors such as age, genetics, lifestyle, chronic inflammation, and stress. Medications used to treat cancer can often lead to cardiotoxicity and the onset of heart failure. Recent studies reveal a close connection between cardiac remodeling, heart failure, and cancer progression. The communication between a remodeled heart and a tumor involves the secretion of various factors, with Periostin consistently elevated in heart failure mouse models.
While in vitro experiments demonstrated the requirement of Periostin for tumor promotion, in vivo studies with mice lacking Periostin following transverse aortic constriction (TAC) surprisingly showed continued tumor growth. This outcome was attributed to elevated levels of cytokines and growth factors in Periostin-deficient mice, many of which are potential Integrin receptor ligands.
To further investigate, cancer cells with a loss-of-function mutation in Integrin β1 were generated. These cells failed to exhibit cardiac remodeling-dependent tumor promotion, suggesting a crucial role for Integrin β1 in this process. Additionally, Renal cell carcinoma cells (Renca) did not respond to factors derived from a failing heart, possibly due to their low basal mRNA levels of Integrin β1.
The study highlights the clinical relevance for cardio-oncology patients with cancers featuring high Integrin β1 levels. Chemotherapy-induced cardiotoxicity in these patients may create a detrimental cycle, impacting prognosis. Understanding the intricate interplay between heart failure, cancer, and specific molecular factors like Periostin and Integrin β1 could offer insights for developing targeted therapies in cardio-oncology. Overall, this is a very intersting study with important findings.
The graphical representation of data in the manuscript needs improvement, specifically in showing individual data points to enhance clarity and transparency. Incorporating a scatter plot or similar visualization would allow readers to better understand the distribution and variability of the results.
Moreover, the Method section requires substantial expansion to include essential details such as the age and gender of the mice used in the experiments. Additionally, critical experimental information, including randomization and blinding procedures, is absent. Addressing these aspects is crucial for ensuring the rigor and reproducibility of the study.
Given that Periostin-positive cells are recognized as perivascular, it is essential to consider the impact of aging on vasculature and perivascular niches. The authors should explicitly discuss and cite these PMID: 33536212, PMID: 36669473 and other studies where ageing is shown to impact perivascular microenvironments whether their own findings have relevance to aging. Insights into how these changes may affect the outcomes of the study would significantly contribute to the overall understanding of the research.
In the Discussion section, it is imperative for the authors to delve into the potential implications of their work on aging and aged mice. A thorough exploration of the relevance of Periostin-positive cells in the context of age-related alterations in the vasculature and perivascular niches should be included. This discussion is pivotal for broadening the scope of the study and addressing potential applications or considerations in the context of aging.
Comments on the Quality of English Language
Minor editing of English language required
Author Response
Heart failure and cancer are two of the deadliest diseases in the Western world, sharing common risk factors such as age, genetics, lifestyle, chronic inflammation, and stress. Medications used to treat cancer can often lead to cardiotoxicity and the onset of heart failure. Recent studies reveal a close connection between cardiac remodeling, heart failure, and cancer progression. The communication between a remodeled heart and a tumor involves the secretion of various factors, with Periostin consistently elevated in heart failure mouse models.
While in vitro experiments demonstrated the requirement of Periostin for tumor promotion, in vivo studies with mice lacking Periostin following transverse aortic constriction (TAC) surprisingly showed continued tumor growth. This outcome was attributed to elevated levels of cytokines and growth factors in Periostin-deficient mice, many of which are potential Integrin receptor ligands.
To further investigate, cancer cells with a loss-of-function mutation in Integrin β1 were generated. These cells failed to exhibit cardiac remodeling-dependent tumor promotion, suggesting a crucial role for Integrin β1 in this process. Additionally, Renal cell carcinoma cells (Renca) did not respond to factors derived from a failing heart, possibly due to their low basal mRNA levels of Integrin β1.
The study highlights the clinical relevance for cardio-oncology patients with cancers featuring high Integrin β1 levels. Chemotherapy-induced cardiotoxicity in these patients may create a detrimental cycle, impacting prognosis. Understanding the intricate interplay between heart failure, cancer, and specific molecular factors like Periostin and Integrin β1 could offer insights for developing targeted therapies in cardio-oncology. Overall, this is a very intersting study with important findings.
- The graphical representation of data in the manuscript needs improvement, specifically in showing individual data points to enhance clarity and transparency. Incorporating a scatter plot or similar visualization would allow readers to better understand the distribution and variability of the results.
Response: Thank you for the comment we noted and resolved it. The graphical representation of mice data was changed. We did not change the graphical representation of culture cell analysis because usually it is represented as a bar graph, we did add the "'n=" of the experiment in the legends.
- Moreover, the Method section requires substantial expansion to include essential details such as the age and gender of the mice used in the experiments. Additionally, critical experimental information, including randomization and blinding procedures, is absent. Addressing these aspects is crucial for ensuring the rigor and reproducibility of the study.
Response: Thank you for the comment we noted and resolved it. We added the relevant information in the graph's legends and the method section.
- Given that Periostin-positive cells are recognized as perivascular, it is essential to consider the impact of aging on vasculature and perivascular niches. The authors should explicitly discuss and cite these PMID: 33536212, PMID: 36669473 and other studies where ageing is shown to impact perivascular microenvironments whether their own findings have relevance to aging. Insights into how these changes may affect the outcomes of the study would significantly contribute to the overall understanding of the research.
- In the Discussion section, it is imperative for the authors to delve into the potential implications of their work on aging and aged mice. A thorough exploration of the relevance of Periostin-positive cells in the context of age-related alterations in the vasculature and perivascular niches should be included. This discussion is pivotal for broadening the scope of the study and addressing potential applications or considerations in the context of aging.
Response: Thank you for the comment we noted and resolved it. We took into consideration your comments and incorporated them into the discussion section.

Round 2
Reviewer 2 Report
Comments and Suggestions for Authors
Authors have addressed my comments and I have no further comments